# Regional Disparities in the Association between Cereal Consumption and Metabolic Syndrome: Results from the China Health and Nutrition Survey

**DOI:** 10.3390/nu11040764

**Published:** 2019-04-01

**Authors:** Lina Huang, Huijun Wang, Zhihong Wang, Jiguo Zhang, Bing Zhang, Gangqiang Ding

**Affiliations:** National Institute for Nutrition and Health, Chinese Center for Disease Control and Prevention, Beijing 100050, China; huangln2999@126.com (L.H.); wanghj@ninh.chinacdc.cn (H.W.); wangzh@ninh.chinacdc.cn (Z.W.); zhangjg@ninh.chinacdc.cn (J.Z.); zhangbing@chinacdc.cn (B.Z.)

**Keywords:** metabolic syndrome, regional disparity, cereals

## Abstract

This study examines regional disparities in the association between cereal consumption and metabolic syndrome (MetS) among Chinese adults. We used data from the longitudinal China Health and Nutrition Survey (CHNS) for 2892 healthy adults aged 18–75 years (1088 in northern China, 1804 in southern China) who had no non-communicable chronic diseases or MetS at the initial visit in 2009 and the follow-up in 2015. We used a 74-item food frequency questionnaire (FFQ) to assess the dietary intake. We defined MetS according to the International Diabetes Federation (IDF) criteria. Multiple logistic regressions stratified by region were performed to estimate the association between cereal consumption and the risk of MetS, and the quantile regression analyzed the relationship between cereal consumption and individual components of MetS in 2015. The rice consumption in southern China (9.00 kg/month) was more than twice that in northern China (3.60 kg/month). Consumption of wheat and wheat products in northern China (4.20 kg/month) was more than twice that in southern China (1.50 kg/month). After we adjusted for potential confounders, rice consumption was inversely associated with a risk of MetS 0.709 (95% CI: 0.458–1.003), the intake of wheat and wheat products was positively associated with a risk of MetS 1.925 (95% CI: 1.292–2.867) in southern China. We found no association between the intake of cereal and the prevalence of MetS in northern China. The quantile regression showed that various cereals were differentially associated with the components of MetS. The association between cereal consumption and the risk of MetS, and the components of MetS varied across these two regions of China.

## 1. Introduction

Cardiovascular diseases (CVDs) are the leading cause of disease burdens and deaths globally [1,2]. In 2015, an estimated 290 million people had CVDs, and it is also the number one cause of mortality in China, accounting for more than 40% of all deaths [3]. The Metabolic syndrome (MetS) is a cluster of metabolic abnormalities that have been associated with an increased risk of developing CVDs and type 2 diabetes mellitus (T2DM) [4,5], along with central obesity, increased blood pressure (BP), increased fasting plasma glucose (FPG), increased triglycerides (TG), and decreased high-density lipoprotein cholesterol (HDL-C). The prevalence of MetS has increased dramatically and has become a serious public health problem in China and worldwide. Based on the findings from the China National Health and Nutrition Surveillance (2010–2012), the overall prevalence rate of MetS among Chinese adults was 11.0% [6].

Various studies have focused on the relationship between cereal consumption and the prevalence of MetS. The literature on rice intake is mixed, with some studies finding a positive relationship [7,8], some no relationship [9,10,11,12,13], and some an inverse relationship [14,15]. In the Japanese population, whole-wheat consumption has been shown to significantly reduce visceral fat obesity, and therefore reduce the risk of MetS [16], but udon noodle consumption was shown to increase the risk of MetS and abdominal obesity [15]. That study also reported that the intake of wheat as a standard high-fiber food might improve the cardiovascular risk profile [17]. Coarse cereals including maize, lentils and legumes, and minor millets, are rich in compounds that help prevent several chronic diseases like CVDs, T2DM, and obesity [18,19]. Several cross-sectional studies among Chinese populations have found different results on the relationship between tuber consumption and the prevalence of MetS [20,21]. In China, rice and wheat are the main staple foods, with regional variations (rice consumption is higher in southern China than in northern China, and wheat consumption is higher in northern China than in southern China). Although cereal consumption in China has decreased over the past three decades, nearly 53.1% of the total caloric intake is still derived from cereals [22]. However, few studies have compared the regional disparities in cereal consumption and the prevalence of MetS in the two regions, especially the prospective study. We used the China Health and Nutrition Survey (CHNS) data to examine the associations between the consumption of individual staple foods and the prevalence of MetS, with attention to regional variations.

## 2. Materials and Methods 

### 2.1. Study Design and Subjects

All data used in this study are from the CHNS, an ongoing large-scale, longitudinal, household-based survey of ten waves (1989–2015). The CHNS originally covered nine provinces in 1989 that varied in demography, geography, economic development, and public resources. It used a multistage, stratified, random cluster process to select communities for the sample. In each community, 20 households were randomly selected and all individuals were surveyed for all the data in each wave. The survey procedure has been described in detail elsewhere [23]. Based on the natural boundary of a line along the Qinling Mountains and the Huaihe River, China is divided into northern and southern geographic regions [24]. The nine provinces selected in the original CHNS are disturbed in these two regions, Jiangsu, Hubei, Hunan, Guangxi and Guizhou in southern China; Heilongjiang, Liaoning, Shandong and Henan in northern China. The survey has only collected fasting blood samples in 2009 and 2015; so, our analysis considered 5132 adults aged 18–75 years who participated in those two waves. Of those participants, we excluded people diagnosed with hypertension, diabetes, cancer or who had MetS in the 2009 wave; pregnant and lactating women; and people with missing diet, BP, waist circumference (WC), or fasting blood data in 2009 or 2015. Hence, our final sample included 2892 people (1088 from northern China, 1804 from southern China). And 728 participants (354 from northern China, 374 from southern China) developed into MetS in 2015.

All participants gave their written informed consent before participating in the study. The study was approved by the Institutional Review Board of the University of North Carolina at Chapel Hill and the National Institute for Nutrition and Health, Chinese Center for Disease Control and Prevention (2015017).

### 2.2. Dietary Data

Our dietary assessment used a semi-quantitative food frequency questionnaire (FFQ) during the baseline visit. The FFQ included 74 food items and nine food categories: cereals, including rice, wheat and wheat products, other staple cereal and tuber; legumes; vegetables; fruits; dairy products; meats, including pork, beef, mutton, poultry, fish and shrimp; eggs; snacks; and alcohol and soft drinks [25]. For each food item, participants reported the frequency of habitual consumption (daily, weekly, monthly, annually or never (the reference category)) and the amount consumed over the past 12 months. Staple foods (e.g., rice, wheat) are consumed daily, but for the episodically consumed (e.g., coarse cereals and tuber) we used monthly consumptions. We converted individual consumption of food items to kilograms per month in the analysis. It was not possible to accurately calculate total energy intake (TEI) from the FFQ, so we used three consecutive 24-h dietary recalls to calculate the TEI.

### 2.3. Definition of MetS

The definition of MetS was based on recommendations from the International Diabetes Federation (IDF) and AHA/NHLBI criteria [26]. According to this definition, a person diagnosed with MetS must meet at least three out of five criteria: (1) Elevated WC (using the Chinese values: ≥90 cm in man, ≥80 cm in female); (2) raised TG (TG ≥ 150 mg/dL (1.7 mmol/L) or drug treatment for elevated triglycerides); (3) reduced HDL-C (<40 mg/dL (1.0 mmol/L) in males, <50 mg/dL (1.3 mmol/L) in females or drug treatment for reduced HDL-C); (4) raised BP (systolic BP [SBP] ≥ 130 mmHg or diastolic BP [DBP] ≥ 85 mmHg or antihypertensive drug treatment in a patient with a history of hypertension); (5) raised FPG (≥100 mg/dL, or drug treatment of elevated glucose).

### 2.4. Anthropomorphic and Blood Chemistry Measurement 

Trained health workers or nurses measured height, weight, WC, and BP following standardized procedures. Height was measured to the nearest 0.1 cm using a height tape (model 206, SECA). With the participant standing and wearing a single layer of clothing, weight was measured to the nearest 0.1 kg using a body fat meter (BC601, TANITA). WC was measured in centimeters at the midway between the lowest rib margin and the top of the iliac crest using a SECA tape measure. BP was measured at least three times using a standard mercury sphygmomanometer after the participants rested at least five minutes in a seated position. SBP was measured at the first appearance of a pulse sound (Korotkoff phase 1) and DBP at the disappearance of the pulse sound (Korotkoff phase 5). We used the mean of three satisfactory measurements for analyses.

Trained nurses collected overnight fasting blood samples. Plasma and serum samples were frozen and stored at −86 degrees centigrade for later laboratory analysis. All samples were analyzed in a national lab in Beijing with strict quality control. FPG was measured by the GOD-PAP (Randox Laboratories Ltd., London, UK). The concentration of serum HDL-C and TG were measured by the enzymatic method and CHOD-PAP (Kyowa Medex Co., Ltd, Tokyo, Japan), respectively [27].

### 2.5. Assessment of Covariates

Trained interviewers used standard questionnaires to collect sociodemographic characteristics, smoking, drinking, physical activity, medical history, annual family income, and community information. We grouped participants into three age categories (18–44, 45–59, and 60–75 years). Marital status was categorized into two statuses (single and married). We calculated the family incomes according to household size and grouped them into low, middle and high. We calculated the community urbanicity index based on 12 multidimensional components reflecting the heterogeneity in economic, social, demographic, and infrastructural characteristics at the community level [28], and grouped the urbanicity into low, middle and high. Physical activity includes four domains: occupational, household chore, leisure time, and transportation activities. Participants reported all activities in average hours per week, and we converted the time spent in each activity into a metabolic equivalent of task (MET) hours per week based on the Compendium of Physical Activities. We categorized the total MET hours per week into low, middle and high. We calculated the body mass index (BMI) as measured weight in kilograms divided by measured height in meters squared, and grouped the BMI into thin, normal and overweight. We classified the smoking status as current or ever/never. We categorized the alcohol drinking status as non/ever-drinking and drinking. We also considered the baseline BMI, baseline WC, BP, TG, FPG, and HDL-C as potential confounders.

### 2.6. Statistical Analysis

Due to regional disparities in dietary intakes, we performed all analyses separately for the northern and southern regions based on statistically significant (*p* < 0.05) interactions between regions and each cereal subtype, and because of geographic variations in cooking and diet in China. For the baseline characteristics of the participants, we expressed categorical variables as numbers (percentages) and examined them with a chi-square test. We expressed continuous variables with normal distribution as means ± standard deviation and used the z-test. We presented skewed distribution variables presented as medians (interquartile ranges) and used non-parametric statistical hypothesis test including the Wilcoxon signed-rank test and the Kruskal-Wallis test. The Cochran and Mantel-Haenszel and Hierarchical Analysis of Variance were used to analyze the baseline characteristics of normal and developed into MetS in 2015 by regions. We grouped the baseline rice, wheat and wheat products, coarse cereal, and tuber consumption into quartiles by region, and constructed a series of multivariable logistic regression models to assess regional disparities in the association of different levels of consumption of each cereal with the prevalence of MetS. We also tested linear trends by assigning median values to quartiles of consumption of each cereal and modeled this variable as a continuous term. After a Box-Cox transformation, the continuous variables still showed a skewed distribution. Therefore, we estimated the association between the consumption of each cereal subtype and the risk of MetS sensitivity factors WC, BP, TG, FPG, and HDL-C with quantile regression models adjusted for baseline covariates. We conducted all statistical analyses using the SAS 9.4 (SAS Institute, Inc., Cary, NC, USA).

## 3. Results

### 3.1. Baseline Characteristics of the Study Population and Cereal Consumption

Of the 2892 participants included in the study, 1088 were northern region (47.24% from men and 52.76% from women), 1804 were southern region (49.28% from men and 50.72% from women). Table 1 shows the baseline characteristics by regions. We found significant differences between the northern and southern regions in age, urbanicity index, physical activity, and BMI. Northern participants were more likely to have higher BMI, WC, and BP (*p* < 0.001). The TEI median was higher among southern participants than among northern (*p* < 0.001). Table 2 shows that the prevalence of MetS in northern China 32.5% was higher than that in southern China 20.7%.

Table 3 shows the baseline characteristics of food consumption according to the medians and interquartile ranges. The monthly consumption of rice in the southern region (9.00 kg/month) was more than twice that in the northern region (3.60 kg/month), and the monthly consumption of wheat and wheat products in northern region (4.20 kg/month) was more than twice that in the southern region (1.50 kg/month). The medians of rice, vegetable and red meat consumption were higher in the southern region (*p* < 0.05), while the medians of wheat and wheat products, coarse cereal, tuber and fruit consumption were higher in the northern region (*p* < 0.001).

### 3.2. The Association between Cereal Subtype Intake and Risk of MetS

Table 4 shows the southern region risk of MetS per quartiles of monthly consumption of each cereal, and Table 5 shows the northern region risk. The risk of MetS decreased as the monthly consumption of rice increased in the southern region. After adjustment for confounding factors (gender; age; marital status; urbanicity index; income level; BMI, smoking; alcohol; physical activity, TEI; vegetable; fruit; red meats), the second and the fourth quartiles of rice consumption were inversely associated with the prevalence of MetS (*p* < 0.05), wheat and wheat products intake were positively associated with the risk of MetS (*p* < 0.05), coarse cereals and tuber intake had no association with the risk of MetS in the southern region. In the northern region, rice, wheat and wheat products, coarse cereals and tuber intake showed no association with the risk of MetS.

### 3.3. The Association between Cereal Subtypes Intake and the Risk of Individual Components of MetS 

Table 6 shows the individual components of the MetS quantile regression results among northern adults. The monthly consumption of rice showed a significant positive association with WC across the entire conditional WC distribution (*p* < 0.05), with the lowest WC association at the 25th quantile, the highest at the 90th quantile (0.197, 0.240, 0.258 and 0.357 at the 25th, 50th, 75th and 90th quantiles, respectively). Rice consumption in the northern region showed a negative association with FPG (−0.036, −0.037 and −0.023 at the 10th, 25th, and 50th quantiles, respectively). The monthly intake of wheat and wheat products was positively associated with FPG and HDL-C, at the lower end of the conditional distribution of FPG (0.010 and 0.012 at the 10th and 50th quantiles, respectively) and HDL-C (0.005 and 0.003 at the 10th, and 25th quantiles, respectively). 

Table 7 shows the individual components of the MetS quantile regression results among southern adults. Monthly consumption of rice was negatively associated with WC, SBP, and HDL-C, and the coefficients were statistically significant at the conditional distribution of WC (−0.105 at the 25th quantile), SBP (−0.259 at the 75th quantile). The wheat and wheat products consumption showed an inverse relationship with FPG across the entire conditional distribution (−0.020, −0.024, −0.028, −0.026 and −0.030 at the 10th, 25th, 50th, 75th, and 90th quantiles, respectively). Coarse cereals consumption had a positive relationship with WC, SBP, FPG and HDL-C, and the coefficients were statistically significant at the conditional distribution of WC (0.836 at the 90th quantile), SBP (0.915, 0.603 at the 10th and 25th quantile, respectively), HDL-C (0.445, 0.53, 0.551 at the 10th, 25th and 50th, respectively). The tuber intake showed an inversely associated with SBP and FPG, and the coefficients were statistically significant at the conditional distribution of FPG (−0.157, −0.105 and −0.076 at the 10th, 25th and 50th, respectively).

## 4. Discussion

Our study involving more than 2892 Chinese individuals without prior MetS or other chronic diseases found rice, wheat and products, coarse cereals and tuber intake disparities in southern and northern China. We found a significant inverse association between rice consumption and MetS, and while it showed a positive association with wheat and wheat products consumption and the prevalence of MetS in southern adults, null associations between each cereal and the risk of MetS were found in northern adults. The associations between cereal subtypes consumption and the risk of the components of MetS varied across the two regions of China.

The association between rice consumption and the prevalence of MetS in southern and northern China is similar to the study from Iran which showed an increased likelihood of T2DM associated with high white rice intake among residents of Tehran (OR: 2.1, 95% CI: 1.1–3.9) and no association in Gorleston [29]. Rice consumption has different effects in different areas. The Korean Genome and Epidemiology Study revealed that the prevalence of MetS was different in men and women with different rice preferences [10]. In Indian and Japan it did not find an association between rice consumption and the prevalence of MetS [9,10,11,13]. The main reason for this result is likely to be that the rice consumption differed between southern China and northern China. A study in Japan, found that a moderate intake of rice might reduce the risks of incident MetS (HRs: 0.83, 95% CI: 0.69–0.99) and high BP (HRs: 0.79, 95% CI: 0.66–0.94) [15]. The Jiangsu Nutrition Study [12], found no significant association between rice intake and incident MetS in a five-year follow-up; rather, the results were in the opposite direction. In the Jiangsu Province in our CHNS southern region, the inconsistency could be partially due to the potential factors. The wheat and wheat products consumption are mostly handmade noodles, steamed bud and handmade dumplings in China, it is much different from western countries. In addition to the wheat and wheat products consumption differed between southern and northern China, the types of wheat and wheat products, which the two regions consumed have also great discrepancy. It mainly consumed the handmade noodles, teamed buns and handmade dumplings in northern China, while people in the southern China mainly consumed processed noodles or pre-packaged pasta. According to the China Food Composition, the sodium content of processed noodle is far more than the handmade noodle [30]. In Japan, wheat consumption reduced the risk of MetS, but the daily intake of wheat products increased the risk of MetS (HRs: 1.19, 95% CI: 1.05–1.35), and abdominal obesity (HRs: 1.15, 95% CI: 1.02–1.29) in urban populations [15]. An intervention study showed that the whole grain wheat bread group decreased its visceral fat area, whereas the refined wheat bread group had no significant changes [16]. Therefore, different processing methods of wheat and wheat products may have a great impact on our results. 

A cross-sectional study of Chinese adults found that high rice consumption was associated with high TG and low HDL-C in the northern region [31]. The studies in Korea and India showed that TG and HDL-C levels were associated with the white rice intake [8,32,33]. It was inconsistent with our result, since the rice intake in northern adult was low in our study. A randomized trial showed that brown rice and white rice can help reduce 24 h glucose and fasting insulin responses among overweight Indians [33]. The present study did not show a significant relationship between coarse cereal consumption and the risk of MetS in either the southern or the northern region. A randomized controlled trial indicated that diabetic groups showed marginal decreases in TG and increases in HDL-C with millet consumption, that is, millet was beneficial for T2DM [34]. More research on the relationship between coarse cereals and lipids, and between glucose and HDL-C are needed. A recent meta-analysis also showed that the consumption of whole grain foods compared to refined foods, improved acutely the postprandial glucose and insulin homeostasis in healthy subjects [35]. Consumption of coarse cereals in China is very low and most consumed do not meet the recommended of dietary guidelines. Few studies have focused on Chinese consumption of coarse cereals, such as millet, corn, and sorghum; so, it is difficult to evaluate the relationship between coarse cereal and the risk of MetS. 

In Western countries studies mainly find a positive correlation between tuber consumption, obesity, and T2DM [36,37,38]. However, our results were inconsistent with it, instead showing an inverse association between tuber consumption and FPG in southern China. This could be explained by the small amount consumed by our participants. A Japanese cohort study found no relationship between low tuber consumption (1–2 times/week) and CVDs, and a positive association between high tuber consumption (≥3 times/week) and CVDs [39]. Another reason for the disparity between China and Western countries could be the different methods of cooking tuber. In China tubers are primarily steamed and boiled, but in Western countries they are mostly fried. The different ways of cooking may affect health differently. Further studies on how cooking methods modify the association between the tuber intake and the risk of MetS are needed.

It suggested that a high consumption of rice might be linked to healthier diet patterns (high fish, seafood and vegetable) in southern China [31]. The inconsistencies of the result between the regions could be due to multiple factors, including different staple food, the amount of cereal consumed, dietary structures, and cooking methods. In addition, a statistical difference in cereal intolerance among the adults in the southern and northern regions of China has also been reported [40]. Our study is the first to examine the association between cereal consumption and MetS in a regional perspective. It mainly studies whether the other food groups and nutrients intake differed between the two regions in the future. This study has several residual confounding factors of an observational study. First, the dietary intake was estimated based on an FFQ that covered the past 12 months of dietary consumption, which may lead to a recall bias, the distinguishability between whole grains and refined grains in our FFQ may not have been clear, and we could not conduct an isocaloric analysis describing results when the total caloric intake is fixed. We used monthly consumptions for episodically consumed foods like coarse cereals and tubers, to relatively stable. Second, measurement errors in the assessments of each cereal consumed are inevitable, and we cannot know the extent to which such an error may bias our results. Third, the sample size of this study is relatively small, containing some of the northern and southern provinces, so we had difficulty generalizing the study results. More studies with larger sample sizes in diverse geographic locations and with longer follow-up periods are needed to examine regional differences.

## 5. Conclusions

In conclusion, our evaluation of the relationship between cereal consumption and the prevalence of MetS and regional disparities in China showed that it had different associations between rice and wheat consumption and a risk of MetS among northern and southern adults. Given the different consumption of staple foods in northern and southern China, regional disparities should be taken into account when studying the relationship between diet and chronic diseases, and making dietary recommendations.

## Figures and Tables

**Table 1 nutrients-11-00764-t001:** Baseline characteristics of the study population by regions.

Factors		Northern	Southern	*p*-Value
Gender, *n* (%)				0.2884
	Men	514(47.24)	889(49.28)	
	Women	574(52.76)	915(50.72)	
Age (%), *n*				
	18–44 years	413(37.96)	631(34.98)	0.0325
	45–59 years	334(30.7)	504(27.94)	
	60–years	341(31.34)	669(37.08)	
Marital status, *n* (%)				0.1024
	Single	86(7.9)	175(9.7)	
	Married	1002(92.1)	1629(90.3)	
Urbanicity index, *n* (%)				
	Low	454(41.73)	506(28.05)	<.0001
	Middle	346(31.8)	621(34.42)	
	High	288(26.47)	677(37.53)	
Income, *n* (%)				0.1180
	Low	353(32.5)	604(33.84)	
	Middle	346(31.86)	611(34.23)	
	High	387(35.64)	570(31.93)	
Physical activity, *n* (%)				0.0009
	Low	331(30.42)	633(35.09)	
	Middle	350(32.17)	615(34.09)	
	High	407(37.41)	556(30.82)	
Smoking, *n* (%)				0.9334
	Ever/Never	723(66.45)	1202(66.63)	
	Current	364(33.46)	601(33.31)	
Alcohol, *n* (%)				0.6489
	Ever/Never	709(65.17)	1158(64.19)	
	Current	379(34.83)	645(35.75)	
BMI, *n* (%)				<0.0001
	Thin	59(5.42)	165(9.15)	
	Normal	750(68.93)	1308(72.51)	
	Overweight	279(25.64)	331(18.35)	
BMI (kg/m^2^)		22.99(21.12,25.07)	22.04(20.28,24.16)	<0.0001
WC (cm)		81(76,87)	79(73,85)	0.0003
SBP (mmHg)		120(111,126)	118(109,126)	0.0003
DBP (mmHg)		80(75,82)	77(70,81)	<0.0001
HDL-C (mg/dL)		54(45,64.5)	55(46,65)	0.0771
TG (mg/dL)		105(73,162)	101(71,151)	0.1422
FPG (mg/dL)		90(83,99)	92(85,100)	0.0457
TEI (kcal/day)		2116.53(1757.5,2519.75)	2247.34(1837.91,2740.79)	<0.0001

Data of categorical variables expressed as number (%); Medians (interquartile ranges) for skewed parameters.

**Table 2 nutrients-11-00764-t002:** Baseline characteristics of normal and developed into MetS in 2015 by regions.

Factors		Northern		Southern		*p*-Value
		Normal	MetS	Normal	MetS	
Case, *n* (%)		734(67.46)	354(32.54)	1430(79.27)	374(20.73)	<0.0001
Gender, *n* (%)						<0.0001
	Men	739(51.68)	150(40.11)	363(49.46)	151(42.66)	
	Women	691(48.32)	224(59.89)	371(50.54)	203(57.34)	
Age, *n* (%)						<0.0001
	18–44 years	543(37.97)	88(23.53)	326(44.41)	87(24.58)	
	45–59 years	385(26.92)	119(31.82)	203(27.66)	131(37.01)	
	60– years	502(35.1)	167(44.65)	205(27.93)	136(38.42)	
Marital status, *n* (%)						0.848
	Single	136(9.51)	39(10.43)	62(8.45)	24(6.78)	
	Married	1294(90.49)	335(89.57)	672(91.55)	330(93.22)	
Urbanicity index, *n* (%)						<0.0001
	Low	421(29.44)	85(22.73)	321(43.73)	133(37.57)	
	Middle	494(34.55)	127(33.96)	238(32.43)	108(30.51)	
	High	515(36.01)	162(43.32)	175(23.84)	113(31.92)	
Income, *n* (%)						0.8334
	Low	482(34.06)	122(32.97)	242(33.06)	111(31.36)	
	Middle	486(34.35)	125(33.78)	221(30.19)	125(35.31)	
	High	447(31.59)	123(33.24)	269(36.75)	118(33.33)	
Physical activity, *n* (%)						0.0007
	Low	484(33.85)	149(39.84)	200(27.25)	131(37.01)	
	Middle	505(35.31)	110(29.41)	237(32.29)	113(31.92)	
	High	441(30.84)	115(30.75)	297(40.46)	110(31.07)	
Smoking, *n* (%)						0.0001
	Ever/Never	927(64.83)	275(73.53)	471(64.17)	252(71.19)	
	Current	502(35.1)	99(26.47)	262(35.69)	102(28.81)	
Alcohol, *n* (%)						0.0162
	Ever/Never	894(62.52)	264(70.59)	476(64.85)	233(65.82)	
	Current	535(37.41)	110(29.41)	258(35.15)	121(34.18)	
BMI, *n* (%)						<0.0001
	Thin	155(10.84)	10(2.67)	54(7.36)	5(1.41)	
	Normal	1101(76.99)	207(55.35)	554(75.48)	196(55.37)	
	Overweight	174(12.17)	157(41.98)	126(17.17)	153(43.22)	
BMI (kg/m^2^)		21.57(19.91,23.49)	24.31(22.27,26.21)	22.2(20.52,24.24)	24.61(22.66,26.55)	<0.0001
WC (cm)		77(71,83)	85(79,90)	80(74,85)	85(79,90)	<0.0001
SBP (mmHg)		117(108,125)	120(113,129)	119(110,125)	121(117,129)	<0.0001
DBP (mmHg)		76(70,81)	79(73,83)	80(72,82)	80(79,83)	<0.0001
HDL-C (mmol/L)		56(47,66)	53(44,61)	55(47,65)	51(43,63)	<0.0001
TG (mmol/L)		96(67,143)	119(82,181)	95(66,146)	125(89,189)	<0.0001
Glucose (mmol/L)		91(84,98)	94.5(88,103)	89(82,98)	94(86,102)	<0.0001

**Table 3 nutrients-11-00764-t003:** Baseline characteristics of monthly food consumption by regions (kg/month).

Subgroups		Northern			Southern		*p*-Value
	Median	P_25th_	P_75th_	Median	P_25th_	P_75th_	
Rice	3.60	0.72	6.00	9.00	6.00	11.70	<0.0001
Wheat and products	4.20	1.77	7.28	1.50	0.56	3.05	<0.0001
Coarse cereals	0.50	0.23	1.04	0.20	0.08	0.50	<0.0001
Tuber	0.80	0.33	1.60	0.30	0.10	0.60	<0.0001
Vegetable	5.73	3.34	8.80	6.00	3.58	9.35	0.043
Fruit	2.40	1.25	4.86	1.78	0.81	3.45	<0.0001
Red meat	0.98	0.49	1.83	1.64	0.85	3.00	<0.0001

Abbreviation: P = percentile.

**Table 4 nutrients-11-00764-t004:** The association between cereal subtype intakes and risk of metabolic syndrome among adults in the southern area.

Subgroups		Q1	Q2	Q3	Q4	*p*-Value
**Rice**						
	Participants	252	603	495	454	
	Median (kg/month)	3.66	6.26	9.19	14.4	
	Model1	Ref	0.716(0.505,1.015)	0.796(0.552,1.147)	0.689(0.470,1.010) *	0.0164
	Model2	Ref	0.611(0.422,0.885) *	0.732(0.499,1.075)	0.646(0.430,0.971) *	0.2841
	Model3	Ref	0.635(0.432,0.934) *	0.744(0.496,1.115)	0.709(0.458,1.003) *	0.4641
**Wheat and products**						
	Participants	450	453	452	449	
	Median (kg/month)	0.23	0.99	2.26	6.42	
	Model1	Ref	1.860(1.310,2.641) *	1.678(1.167,2.412) *	1.979(1.379,2.841) *	0.0449
	Model2	Ref	1.641(1.131,2.381) *	1.496(1.020,2.195) *	1.800(1.231,2.632) *	0.0130
	Model3	Ref	1.601(1.092,2.346) *	1.479(0.997,2.192) *	1.925(1.292,2.867) *	0.0146
**Coarse cereals**						
	Participants	449	476	435	444	
	Median (kg/month)	0.02	0.14	0.34	1.72	
	Model1	Ref	0.934(0.664,1.314)	1.199(0.853,1.685)	1.184(0.837,1.676)	0.2862
	Model2	Ref	1.011(0.705,1.451)	1.225(0.852,1.761)	1.257(0.869,1.819)	0.3987
	Model3	Ref	1.015(0.700,1.471)	1.185(0.815,1.722)	1.283(0.866,1.899)	0.2467
**Tuber**						
	Participants	379	513	434	478	
	Median (kg/month)	0.02	0.16	0.38	1.29	
	Model1	Ref	0.989(0.704,1.388)	0.767(0.534,1.100)	0.837(0.589,1.191)	0.6631
	Model2	Ref	0.924(0.646,1.322)	0.745(0.510,1.088)	0.834(0.576,1.207)	0.4796
	Model3	Ref	0.979(0.677,1.415)	0.809(0.549,1.192)	0.826(0.561,1.216)	0.4168

Abbreviation: Q = quarter. Data intake expressed as median (25th percentile, 75th percentile); Model1: Crude; Model2: adjusted gender, age, marital status, income level, urbanicity index; Model3: model 2+body mass index, smoking, alcohol, physical activity, TEI, vegetable, fruit, red meat consumption, and other type of cereals intake. * *p* < 0.05.

**Table 5 nutrients-11-00764-t005:** The association between cereal subtype intakes and risk of metabolic syndrome among adults in northern area.

Subgroups		Q1	Q2	Q3	Q4	*p*-Value
**Rice**						
	Participants	271	274	283	260	
	Median (kg/month)	0.36	2.01	5.44	11.77	
	Model1	Ref	0.789(0.539,1.153)	0.933(0.625,1.393)	0.998(0.649,1.535)	0.5702
	Model2	Ref	0.779(0.520,1.166)	0.966(0.631,1.480)	1.059(0.669,1.676)	0.3811
	Model3	Ref	0.690(0.457,1.042)	0.900(0.583,1.389)	0.981(0.610,1.578)	0.5617
**Wheat and products**						
	Participants	272	274	270	272	
	Median (kg/month)	1.04	2.97	5.71	13.75	
	Model1	Ref	0.974(0.663,1.431)	1.006(0.681,1.487)	1.073(0.705,1.635)	0.5198
	Model2	Ref	0.929(0.617,1.400)	0.949(0.625,1.441)	1.008(0.642,1.582)	0.7345
	Model3	Ref	0.930(0.614,1.410)	0.964(0.631,1.473)	0.918(0.576,1.464)	0.7887
**Coarse cereals**						
	Participants	272	278	267	271	
	Median (kg/month)	0.10	0.37	0.74	3.84	
	Model1	Ref	1.049(0.728,1.510)	0.755(0.518,1.100)	0.797(0.543,1.170)	0.1416
	Model2	Ref	1.046(0.707,1.548)	0.764(0.510,1.145)	0.780(0.518,1.175)	0.1350
	Model3	Ref	1.059(0.712,1.576)	0.756(0.503,1.137)	0.721(0.469,1.109)	0.0422
**Tuber**						
	Participants	272	237	344	235	
	Median (kg/month)	0.14	0.48	1.16	3.68	
	Model1	Ref	0.877(0.601,1.278)	0.704(0.494,1.004)	0.724(0.483,1.084)	0.1556
	Model2	Ref	0.910(0.608,1.364)	0.801(0.549,1.167)	0.787(0.507,1.222)	0.3035
	Model3	Ref	0.843(0.559,1.273)	0.758(0.516,1.115)	0.724(0.461,1.139)	0.2137

Abbreviation: Q = quarter. Data intake expressed as median (25th percentile, 75th percentile); Model1: Crude; Model2: adjusted gender, age, marital status, income level, urbanicity index; Model3: model 2+body mass index, smoking, alcohol, physical activity, TEI, vegetable, fruit, red meat consumption and other type of cereals intake.

**Table 6 nutrients-11-00764-t006:** Coefficient estimates from a quantile regression on individual components of MetS among northern China by cereal subtype intake.

Variable				Quantile ^#^		
		10th	25th	50th	75th	90th
WC						
	Rice	0.110 (−0.079,0.298)	0.197 (0.047,0.347) *	0.240 (0.071,0.409) *	0.258 (0.091,0.425) *	0.357 (0.127,0.586) *
	Wheat and products	−0.004 (−0.210,0.202)	−0.016 (−0.124,0.091)	−0.027 (−0.138,0.085)	0.030 (−0.105,0.165)	0.160 (−0.038,0.358)
	Coarse cereals	0.005 (−0.242,0.251)	−0.033 (−0.191,0.124)	−0.056 (−0.256,0.144)	−0.066 (−0.313,0.181)	−0.102 (−0.418,0.215)
	Tuber	0.236 (−0.232,0.704)	−0.026 (−0.362,0.309)	0.177 (−0.186,0.539)	0.066 (−0.341,0.473)	−0.057 (−0.733,0.619)
SBP						
	Rice	0.116 (−0.188,0.421)	0.082 (−0.248,0.412)	0.060 (−0.200,0.320)	0.115 (−0.210,0.440)	0.144 (−0.327,0.615)
	Wheat and products	−0.084 (−0.306,0.137)	−0.010 (−0.214,0.193)	−0.104 (−0.248,0.040)	−0.072 (−0.363,0.219)	0.206 (−0.282,0.695)
	Coarse cereals	0.090 (−0.405,0.585)	0.069 (−0.295,0.434)	0.046 (−0.230,0.322)	−0.004 (−0.316,0.308)	−0.050 (−0.489,0.388)
	Tuber	−0.785 (−1.920,0.351)	−0.439 (−1.349,0.471)	−0.455 (−1.216,0.307)	−0.600 (−1.336,0.135)	−0.012 (−1.686,1.662)
TG						
	Rice	0.005 (−0.001,0.011)	0.005 (−0.002,0.013)	0.002 (−0.007,0.012)	0.000 (−0.014,0.014)	0.012 (−0.020,0.043)
	Wheat and products	0.002 (−0.003,0.007)	−0.001 (−0.007,0.004)	−0.005 (−0.013,0.004)	−0.006 (−0.014,0.001)	−0.013 (−0.023,−0.002) *
	Coarse cereals	0.000 (−0.009,0.008)	0.000 (−0.008,0.007)	0.000 (−0.010,0.010)	−0.001 (−0.016,0.014)	−0.005 (−0.058,0.049)
	Tuber	0.012 (0.001,0.024) *	0.007 (−0.009,0.023)	−0.001 (−0.021,0.019)	−0.004 (−0.037,0.028)	−0.031 (−0.082,0.019)
FPG						
	Rice	−0.036 (−0.053,−0.018) *	−0.037 (−0.053,−0.021) *	−0.023 (−0.035,−0.012) *	−0.007 (−0.023,0.009)	0.015 (−0.010,0.039)
	Wheat and products	0.010 (0.004,0.016) *	0.007 (−0.002,0.015)	0.012 (0.004,0.020) *	0.012 (−0.001,0.025)	0.024 (−0.008,0.055)
	Coarse cereals	0.009 (−0.018,0.035)	0.008 (−0.011,0.027)	0.004 (−0.011,0.019)	0.000 (−0.024,0.024)	−0.007 (−0.054,0.042)
	Tuber	0.007 (−0.042,0.057)	−0.006 (−0.042,0.03)	−0.004 (−0.024,0.016)	−0.004 (−0.047,0.039)	0.032 (−0.053,0.117)
HDL-C						
	Rice	−0.002 (−0.010,0.006)	−0.001 (−0.006,0.004)	−0.002 (−0.006,0.002)	−0.002 (−0.007,0.003)	0.000 (−0.007,0.007)
	Wheat and products	0.005 (0.001,0.009) *	0.003 (0.000,0.007) *	0.002 (−0.001,0.004)	0.003 (−0.002,0.008)	0.004 (−0.004,0.011)
	Coarse cereals	0.003 (−0.018,0.023)	0.002 (−0.004,0.007)	0.001 (−0.004,0.006)	0.001 (−0.006,0.007)	−0.001 (−0.016,0.015)
	Tuber	−0.005 (−0.024,0.015)	−0.002 (−0.016,0.012)	−0.001 (−0.014,0.011)	0.005 (−0.008,0.019)	0.014 (−0.003,0.031)

Adjusted gender, age, marital status, income level, urbanicity index, physical activity, drinking, smoking, baseline value of BMI and each homologous MetS component, TEI, vegetable, fruit, red meat consumption, and other type of cereals intake. ^#^ Coefficient (95% CI); * *p* < 0.05.

**Table 7 nutrients-11-00764-t007:** Coefficient estimates from a quantile regression on individual components of MetS among southern China by cereals consumption.

Variable				Quantile ^#^		
		10th	25th	50th	75th	90th
WC						
	Rice	−0.011 (−0.168,0.146)	−0.105 (−0.203,−0.008) *	−0.062 (−0.185,0.06)	−0.067 (−0.183,0.049)	−0.087 (−0.237,0.063)
	Wheat and products	−0.059 (−0.203,0.085)	−0.018 (−0.122,0.086)	0.033 (−0.074,0.139)	0.215 (0.024,0.406) *	0.161 (−0.158,0.481)
	Coarse cereals	0.271 (−0.119,0.660)	0.256 (−0.078,0.589)	0.197 (−0.162,0.557)	0.140 (−0.518,0.799)	0.836 (0.065,1.608) *
	Tuber	−0.187 (−0.724,0.350)	−0.232 (−0.920,0.457)	−0.019 (−0.583,0.545)	0.174 (−0.611,0.959)	0.363 (−0.607,1.333)
SBP						
	Rice	−0.025 (−0.259,0.209)	−0.056 (−0.268,0.156)	−0.038 (−0.24,0.163)	−0.209 (−0.437,0.018)	−0.254 (−0.608,0.100)
	Wheat and products	−0.083 (−0.324,0.158)	−0.176 (−0.440,0.089)	−0.275 (−0.586,0.036)	0.045 (−0.280,0.369)	0.030 (−0.424,0.484)
	Coarse cereals	0.915 (0.422,1.407) *	0.603 (0.155,1.051) *	0.054 (−0.506,0.614)	−0.155 (−1.264,0.955)	−0.084 (−1.638,1.471)
	Tuber	−0.516 (−2.584,−0.448) *	−0.354 (−1.935,1.226)	0.335 (−0.841,1.511)	−0.136 (−1.341,1.069)	−0.648 (−2.408,1.113)
TG						
	Rice	−0.001 (−0.006,0.004)	0.001 (−0.005,0.007)	−0.001 (−0.009,0.007)	0.003 (−0.012,0.018)	−0.001 (−0.022,0.019)
	Wheat and products	−0.005 (−0.012,0.003)	−0.005 (−0.011,0.002)	−0.005 (−0.015,0.005)	−0.008 (−0.024,0.008)	−0.007 (−0.032,0.017)
	Coarse cereals	0.001 (−0.031,0.032)	0.003 (−0.026,0.033)	−0.010 (−0.040,0.019)	−0.004 (−0.046,0.038)	−0.037 (−0.124,0.050)
	Tuber	0.024 (−0.019,0.067)	0.019(−0.029,0.066)	0.036 (−0.009,0.081)	0.016 (−0.049,0.082)	−0.047 (−0.147,0.053)
FPG						
	Rice	0.013 (0.002,0.023) *	0.004 (−0.005,0.013)	−0.006 (−0.016,0.005)	−0.002 (−0.017,0.013)	−0.008 (−0.034,0.019)
	Wheat and products	−0.020 (−0.039,0.000) *	−0.024 (−0.039,−0.01) *	−0.028 (−0.043,−0.014) *	−0.026 (−0.043,−0.008) *	−0.030 (−0.050,−0.011) *
	Coarse cereals	0.053 (−0.005,0.111)	0.025 (−0.015,0.064)	0.000 (−0.030,0.031)	−0.005 (−0.045,0.036)	−0.005 (−0.072,0.061)
	Tuber	−0.157 (−0.259,−0.054) *	−0.105 (−0.174,−0.035) *	−0.076 (−0.128,−0.023) *	−0.041 (−0.126,0.043)	−0.029 (−0.140,0.083)
HDL-C						
	Rice	−0.084 (−0.256,0.089)	−0.037 (−0.165,0.091)	−0.094 (−0.219,0.032)	−0.087 (−0.237,0.063)	−0.013 (−0.183,0.157)
	Wheat and products	0.035 (−0.147,0.218)	0.148 (−0.026,0.322)	0.109 (−0.055,0.273)	0.180 (−0.043,0.403)	0.169 (−0.147,0.486)
	Coarse cereals	0.445 (0.006,0.883) *	0.530 (0.125,0.934) *	0.551 (0.108,0.993) *	0.593 (−0.036,1.222)	0.856 (−0.009,1.720)
	Tuber	0.324 (−0.344,0.992)	0.142 (−0.653,0.937)	−0.077 (−0.715,0.561)	0.358 (−0.860,1.576)	0.901 (−0.147,1.949)

Adjusted gender, age, marital status, income level, urbanicity index, physical activity, drinking, smoking, baseline value of BMI and each homologous MetS component, TEI, vegetable, fruit, red meat consumption, and other type of cereals intake. ^#^ Coefficient (95%CI); * *p* < 0.05.

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
