# Peer review of "Regional Disparities in the Association between Cereal Consumption and Metabolic Syndrome: Results from the China Health and Nutrition Survey"

_nutrients, 2019, doi:10.3390/nu11040764_

Round 1

Reviewer 1 Report

This manuscript describes the results for the association between cereals consumption and metabolic syndrome based on data from the longitudinal China Health and Nutrition Survey. Overall, this paper contributes to the field, but there some points that need being clarified and examined further.  - A main concern was that outdated definition of metabolic syndrome was used in this study. A primary analysis should be based on a harmonized definition of metabolic syndrome (Circulation 2009;120;1640-1645). - Number of subjects used in assessing prevalence and incidence of metabolic syndrome should be different. But, there is no information about this aspect at the section of study design and subjects. It is very confusing that the authors removed the participants with metabolic syndrome at baseline, but analyzed the prevalence of metabolic syndrome. - The number of subjects should be presented in each quartile in tables. - In the association between each cereal and risk of metabolic syndrome, consumption of other type of cereals should be considered in the model.

Author Response

Point 1: This manuscript describes the results for the association between cereals consumption and metabolic syndrome based on data from the longitudinal China Health and Nutrition Survey. Overall, this paper contributes to the field, but there some points that need being clarified and examined further.

A main concern was that outdated definition of metabolic syndrome was used in this study. A primary analysis should be based on a harmonized definition of metabolic syndrome (Circulation 2009;120;1640-1645).

Response 1: Thank you for your comment. We carefully considered which definition to adopt. Taking into account comparison between relevant studies, we chose the old version. As you suggested, we use the newly harmonized definition of metabolic syndrome (Circulation 2009;120;1640-1645) to update the analysis results in our revised manuscript.

Point 2: Number of subjects used in assessing prevalence and incidence of metabolic syndrome should be different. But there is no information about this aspect at the section of study design and subjects. It is very confusing that the authors removed the participants with metabolic syndrome at baseline, but analysed the prevalence of metabolic syndrome.

Response 2: Thank you for your comment and it is worth discussing. We excluded the subjects with metabolic syndrome at baseline, so we could only give a description of metabolic syndrome prevalence, and we make it clear in method and the result section.

 In addition, we removed those who were found to have metabolic syndrome, hypertension, diabetes and cancers at baseline was according the conditions based on prospective study design, sample with the outcomes in baseline should be excluded. And excluded the subjects with cancer, for the possibility of change in their dietary behavior. If we don’t exclude these subjects, it may have inverted effects or weaken the effects.

Point 3: The number of subjects should be presented in each quartile in tables.

Response 3: We added the number of subjects in each quartile in tables (Table 3 and Table4) as you suggested.

Point 4: In the association between each cereal and risk of metabolic syndrome, consumption of other type of cereals should be considered in the model.

Response 4: Thank you for your comment. It is one important issue. We did adjust for the other type of cereals when we analyzed the association between any one type of cereal and risk of Met S, but we forgot mentioning it in the footnote of the tables. We added this information now.

Reviewer 2 Report

Revision:

please add in Table 1 and Table 2 total energy intake/day;

please add another model of adjustment for all the performed analysis including total energy intake as covariate, please add the description and discussion of obtained results;

please implement the results from a recent meta-analysis [PMID: 28753929], which demonstrated that the consumption of whole-grains improves acutely the postprandial glucose and insulin homeostasis when compared to similar refined foods in healthy subjects.

Author Response

Point 1:

-please add in Table 1 and Table 2 total energy intake/day;

-please add another model of adjustment for all the performed analysis including total energy intake as covariate, please add the description and discussion of obtained results;

-please implement the results from a recent meta-analysis [PMID: 28753929], which demonstrated that the consumption of whole-grains improves acutely the postprandial glucose and insulin homeostasis when compared to similar refined foods in healthy subjects.

Response 1: Thank you for your advices, and they are crucial issues. In our study, we used the semi-quantitative food frequency questionnaire (FFQ) to assess the intakes of foods or food groups. Given that it is hard to calculate total energy intake (TEI) accurately based on the FFQ, we used data from consecutive 3-d 24-h recalls to calculate TEI and adjusted for the TEI in the following analyses.

Thanks for your great input of a recent meta-analysis [PMID: 28753929], we added evidence-based relevant points in the discussion part.

Reviewer 3 Report

It is not clear whether the relationship between metabolic syndrome and intake is based on the relationship between baseline intake and subsequent incidence of metabolic syndrome or whether the comparison is with metabolic at baseline. The use of English makes the paper difficult to follow.

Metabolic syndrome is defined on the present of three of the following of obesity (either waist circumference above cut=off or BMI>30 kg/m2), dyslipidemia (low HDL/raised TAG), hypertension and raised fasting glucose.

It would seem repetitive to look at relationships between the components of metabolic syndrome

Differences in the age structure of between the North and South could explain some of the differences.

Author Response

Point 1: It is not clear whether the relationship between metabolic syndrome and intake is based on the relationship between baseline intake and subsequent incidence of metabolic syndrome or whether the comparison is with metabolic at baseline. The use of English makes the paper difficult to follow.

Response 1: Thank you for comment. We aimed to examine regional disparity in the relationship between cereals intake at baseline and incidence of metabolic syndrome. We clarified this point in the whole text. The manuscript has been edited by a native English speaker.

Point 2: Metabolic syndrome is defined on the present of three of the following of obesity (either waist circumference above cut=off or BMI>30 kg/m2), dyslipidemia (low HDL/raised TAG), hypertension and raised fasting glucose.

Response 2: Thank you for your important reminding. Indeed, we failed to follow up the update of IDF definition in time. We use the definition of metabolic syndrome based on a harmonized definition of metabolic syndrome (Circulation 2009;120;1640-1645) in our revised manuscript. The definition of MetS was based on recommendations from IDF and AHA/NHLBI criteria. According to the definition, for a person to be defined as having the MetS they must meet at least three out of five criteria:

(1) Elevated waist circumference (WC) (Chinese values: 90 cm in man, 80 cm in female); (2) raised triglycerides (TG) (TG150 mg/dL (1.7 mmol/L) or drug treatment for elevated triglycerides);

(3) reduced high density lipoprotein-cholesterol (HDL-C) (< 40 mg/dL (1.0 mmol/L) in males, < 50 mg/dL (1.3 mmol/L) in females or drug treatment for reduced HDL-C);

(4) raised blood pressure (BP) (systolic BP 130 mmHg or diastolic BP 85 mmHg or antihypertensive drug treatment in a patient with a history of hypertension);

(5) raised fasting plasma glucose (FPG) (≥ 100 mg/dL, or drug treatment of elevated glucose).

Point 3: It would seem repetitive to look at relationships between the components of metabolic syndrome

Response 3: Each of the components of MetS definition is an important cardiometabolic risk factor. We’d like to get specific results focusing on the associations between the intakes of cereal and sub-type cereals and each component of MetS beyond cereals intake-MetS relationship, which contributes to comprehensive understanding.

Point 4: It would seem repetitive to look at relationships between the components of metabolic syndrome

Response 4: Based on the statistically significant interaction between cereal intakes and geographic region on MetS risk, we performed separate regression modelling adjusting for age stratified by geographic region. As you suggested, we tested the age structure of the North and South and found no significant difference between their age structures.

Reviewer 4 Report

The manuscript "Regional disparity in association between cereals consumption..." aims at investigating the correlation between cereal consumption and metabolic syndrome, and how this correlation varies in different regions of China (i.e. Southern vs Northern).

The topic and the work done by the research group are interesting, but unfortunately the manuscript cannot be published in the present form. At this stage, the linguistic serious flaws hamper the reader to follow the text. I'm sure the background, the approach, the aims and outcomes of the study can be valorized.

My suggestion is to rewrite the manuscript, supported by a fluent English speaker, and resubmit the paper for consideration.

Some general comments:

- use acronyms only after you specified what they stand for;

- tables: change the layout. Please, find an alternative to the use of numbers into brackets.

Author Response

Point 1: The manuscript "Regional disparity in association between cereals consumption..." aims at investigating the correlation between cereal consumption and metabolic syndrome, and how this correlation varies in different regions of China (i.e. Southern vs Northern).

Response 1: thank you for your comment, we really didn’t make it clear in our study. There were significant differences in cereals intake at baseline between north and south. The consumption of rice in southern(9.00kg/month) was more than twice that in northern (3.60 kg/month). Inversely, the consumption of wheat and products in northern (4.20 kg/month) was also more than twice in southern(1.53kg/month). And we attempted to examine the relationship between these differences and the subsequent incidence of metabolic syndrome.

Point 2: The topic and the work done by the research group are interesting, but unfortunately the manuscript cannot be published in the present form. At this stage, the linguistic serious flaws hamper the reader to follow the text. I'm sure the background, the approach, the aims and outcomes of the study can be valorised.

My suggestion is to rewrite the manuscript, supported by a fluent English speaker, and resubmit the paper for consideration.

Some general comments:

- use acronyms only after you specified what they stand for;

- tables: change the layout. Please, find an alternative to the use of numbers into brackets.

Response 2: Thank you for your comment. The manuscript is edited by a native English speaker before submission. We revised the use of acronyms and the layout of tables.

Round 2

Reviewer 2 Report

no further comments

Author Response

Thank you very much.

Reviewer 3 Report

A key item is missing is the incidence of metabolic syndrome on follow-up in the abstract. Although, the incidence is briefly referred to in the text,  it is central to analysis. It would be preferable  to have a table showing the characteristics of those who developed developed metabolic syndrome compared with those who did not in North and South China. 

The analysis of the component of metabolic syndrome in tables 5 and 6 is superfluous and could be removed from the manuscript.

As hypertension is a major component of metabolic syndrome, is there any evidence that salt intakes are higher in Northern than Southern China? Climatic differences also need to be considered as ambient temperature affects blood pressure.

A further issue is whether the percentage energy from protein, carbohydrate and fat differed between Northern and Southern China. Although meat intake was higher in the South, wheat contains more protein than rice. The paper as it stands is difficult for the reader to assess nutrient composition. It would also be helpful to describe how the wheat is consumed in Northern China as noodles, bulgur or bread.

In terms of presentation of results in the abstract it would be better to refer to hazards per inter quintile range ie top quintile versus bottom quintile. 

There are also a number of typos that need correct. 

The numbers in the text for the prevalence of metabolic syndrome 32.5% in Northern China and 20.7% in Southern China do not tally with the Figure 1 please check. If the figure is used it should use age and gender adjusted prevalence.

Author Response

Point 1: A key item is missing is the incidence of metabolic syndrome on follow-up in the abstract. Although, the incidence is briefly referred to in the text, it is central to analysis. It would be preferable to have a table showing the characteristics of those who developed metabolic syndrome compared with those who did not in North and South China.

Response 1: Thank you for suggestions. We had added the results of the characteristics of those who developed metabolic syndrome and those who did not in North and South China.

Point 2: The analysis of the component of metabolic syndrome in tables 5 and 6 is superfluous and could be removed from the manuscript.

Response 2: Thank you for suggestions. Each of the components of MetS definition is an important cardiometabolic risk factor. We’d like to get specific results focusing on the associations between the intakes of cereal and each component of MetS beyond cereals intake-MetS relationship, which contributes to comprehensive understanding. So, we want to show it in our manuscript.

Point 3: As hypertension is a major component of metabolic syndrome, is there any evidence that salt intakes are higher in Northern than Southern China? Climatic differences also need to be considered as ambient temperature affects blood pressure.

Response 3: Thank you for your reminding, and they are crucial issues. In our study, we used the semi-quantitative food frequency questionnaire (FFQ) to assess the intakes of foods or food groups. The foods or food groups do not include salt, so we cannot get any information of salt intake in northern and southern China by FFQ. The climate might actually affect blood pressure, we have considered the comprehensive conditions (include climate) of the 2 regions when we divided into northern and southern China.

Point 4: A further issue is whether the percentage energy from protein, carbohydrate and fat differed between Northern and Southern China. Although meat intake was higher in the South, wheat contains more protein than rice. The paper as it stands is difficult for the reader to assess nutrient composition. It would also be helpful to describe how the wheat is consumed in Northern China as noodles, bulgur or bread.

Response 4: Your suggestion is very important. Our study was mainly focus on foods or food groups. Nutrients intake differed between the northern and Southern China are also our major research in the future. We make varieties of wheat (e.g. handmade noodle, steamed bread) consumed in the 2 regions clearly in the revised manuscript.

Point 5: In terms of presentation of results in the abstract it would be better to refer to hazards per inter quintile range ie top quintile versus bottom quintile.

Response 5: Thank you your suggestion, we made revision in the revised manuscript.

Point 6: There are also a number of typos that need correct.

Response 6: We made revision in the revised manuscript.

Point 7: The numbers in the text for the prevalence of metabolic syndrome 32.5% in Northern China and 20.7% in Southern China do not tally with the Figure 1 please check. If the figure is used it should use age and gender adjusted prevalence.

Response 7:  The Figure 1 is the result of our first manuscript, the prevalence of metabolic syndrome 32.5% in Northern China and 20.7% in Southern China is the results of reanalysis according to the reviewer’s suggestion of using the update metabolic syndrome definition by IDF and AHAA/NHLBI, so in our first revised manuscript we had deleted the Figure 1.

Reviewer 4 Report

Dear Authors,

Thank you for amending your manuscript. The English language was improved and the value of your work is more evident. Several mistakes are however still present within the text, so it requires further revision from a linguistic point of view.

The “Results” section is too long with respect to “Discussion”.

Moreover, after line 1376 you rather discuss the strengths and limitations of your work.

So, as a general comment, I would suggest

-          having one single section for “Results and Discussion”,

-          moving the content currently reported in lines 1376-1393 to a new section on limitations and strengths of the study, and

-          having a section for “Conclusions” which should be more elaborated than it currently is.

Line 1203. Please, specify which studies are consistent with yours. Add reference.

Author Response

Point 1: Thank you for amending your manuscript. The English language was improved and the value of your work is more evident. Several mistakes are however still present within the text, so it requires further revision from a linguistic point of view.

The “Results” section is too long with respect to “Discussion”.

Moreover, after line 1376 you rather discuss the strengths and limitations of your work.

So, as a general comment, I would suggest

- having one single section for “Results and Discussion”,

- moving the content currently reported in lines 1376-1393 to a new section on limitations

and strengths of the study, and

- having a section for “Conclusions” which should be more elaborated than it currently is. Line 1203. Please, specify which studies are consistent with yours. Add reference.

Response 1: Thank you for your valuable suggestions. According to your suggestions the section of discussion and conclusions have been made a major revision.